# Sub-wavelength modulation of $\chi^{(2)}$ optical nonlinearity in organic thin films

Yixin Yan[1], Yakun Yuan[2], Baomin Wang[1], Venkatraman Gopalan[2] & Noel C. Giebink[1]

Modulating the second-order nonlinear optical susceptibility ($\chi^{(2)}$) of materials at the nanoscale represents an ongoing technological challenge for a variety of integrated frequency conversion and nonlinear nanophotonic applications. Here we exploit the large hyperpolarizability of intermolecular charge transfer states, naturally aligned at an organic semiconductor donor–acceptor (DA) interface, as a means to control the magnitude and sign of $\chi^{(2)}$ at the nanoscale. Focusing initially on a single pentacene-$C_{60}$ DA interface, we confirm that the charge transfer transition is strongly aligned orthogonal to the heterojunction and find that it is responsible for a large interfacial nonlinearity probed via second harmonic generation that is sufficient to achieve $d_{33} > 10\,\mathrm{pm\,V^{-1}}$, when incorporated in a non-centrosymmetric DA multilayer stack. Using grating-shadowed oblique-angle deposition to laterally structure the DA interface distribution in such multilayers subsequently enables the demonstration of a $\chi^{(2)}$ grating with 280 nm periodicity, which is the shortest reported to date.

[1] Department of Electrical Engineering, The Pennsylvania State University, University Park, Pennsylvania 16802, USA. [2] Department of Materials Science and Engineering, The Pennsylvania State University, University Park, Pennsylvania 16802, USA. Correspondence and requests for materials should be addressed to N.C.G. (email: ncg2@psu.edu).

Organic thin films have long been pursued for nonlinear optical (NLO) applications such as frequency conversion[1–3], electro-optic modulation[4–6], spectroscopy and imaging[7,8] because they combine large and fast nonlinear responses together with versatile processing and integration capabilities[9–13]. Many of these functionalities rely on the second-order NLO susceptibility, $\chi^{(2)}$, which in turn depends on high dipole moment charge transfer (CT) transitions that maximize the (first) molecular hyperpolarizability, $\beta$ (refs 9,12,14). These CT transitions are traditionally intramolecular in nature, driven between electron donor (D) and acceptor (A) moieties on individual NLO molecules that are electrically poled into alignment within a glassy polymer matrix to break inversion symmetry[2,9,13,15].

Intermolecular CT states existing at the interface between separate donor and acceptor molecules provide an alternate basis for achieving large $\beta$. This possibility was explored early on in theoretical work by Di Bella et al.[16] and was subsequently pursued in experiments by Kajzar and co-workers[17,18], who thermally evaporated repeating donor–acceptor–spacer (DAS) multilayers based on fullerene $C_{60}$ and various small molecule donors to create non-centrosymmetric DAS...DAS stacks. Despite modest nonlinear coefficients $d \sim 5\,\mathrm{pm\,V^{-1}}$ (at fundamental

$\chi^{(2)}$ modulation

**Figure 1 | Generation and modulation of $\chi^{(2)}$ by intermolecular charge transfer states.** Stacking pairs of organic donor (D, red) and acceptor (A, blue) thin films separated by insulating spacer (S, grey) layers leads to a high density of naturally aligned, intermolecular DA charge transfer states (represented by the dashed ovals), resulting in the creation of a non-centrosymmetric, $\chi^{(2)}$-active composite material. Grating-shadowed oblique-angle deposition illustrated in the lower portion of the figure provides a simple way to periodically reverse the DA interface orientation, enabling the creation of $\chi^{(2)}$ gratings for quasi-phase matching with arbitrarily high spatial frequency, $\Lambda$.

wavelength $\lambda_\omega = 1{,}064\,\mathrm{nm}$), these authors nevertheless demonstrated the viability of this approach through increased second harmonic generation (SHG) relative to centrosymmetric DA blend control films.

More broadly, a long-standing challenge in the field for organic and inorganic NLO materials alike lies in modulating $\chi^{(2)}$ at short ($<1\,\mu\mathrm{m}$) length scales. This capability is technologically important for backward-wave quasi-phase matching that is key to a variety of integrated frequency conversion applications, such as mirrorless optical parametric oscillation[19–21], yet it remains a major challenge for conventional periodic poling methods (which are typically limited to periods $>2\,\mu\mathrm{m}$)[19,22,23]. By contrast, the ability to modulate $\chi^{(2)}$ at short periods emerges naturally for an interfacial CT nonlinearity since the DA interface orientation can be structured using standard nanofabrication techniques. In this context, the primary motivation for exploring such interfacial CT nonlinear materials centres not on the absolute nonlinearity they achieve, but rather on the unique opportunity they provide to modulate $\chi^{(2)}$ at sub-micrometer periods.

Here we firmly establish the contribution of intermolecular CT states to the $\chi^{(2)}$ response in this class of organic NLO materials, using the prototypical organic semiconductor donors pentacene and rubrene paired with the acceptor $C_{60}$. Through incidence angle and polarization-dependent SHG measurements, we determine effective NLO bulk $d_{33}$ coefficients $>10\,\mathrm{pm\,V^{-1}}$ at a fundamental wavelength $\lambda_\omega = 800\,\mathrm{nm}$. The origin of this nonlinearity is understood from the electronic properties and orientational order parameter of the interfacial CT state determined from polarized absorption and electroabsorption (EA) measurements of the DA interface. We go on to demonstrate 280 nm period modulation of $\chi^{(2)}$ via backward-wave SH diffraction from a periodic DAS multilayer fabricated by oblique-angle deposition on a grating. These results open up a route to engineer organic semiconducting thin films with a substantial $\chi^{(2)}$ nonlinearity that can be modulated at shorter length scales than has previously been possible.

## Results

**$\chi^{(2)}$ nonlinearity from intermolecular CT states.** The hyperpolarizability associated with a given CT transition can be qualitatively understood from a simple two-level model[24,25], which predicts:

$$\beta_{CT}(2\omega;\omega,\omega) = \frac{3q^2}{2\hbar} \frac{\Delta\mu_{CT} f_{CT} E_{CT}}{\left[E_{CT}^2 - (\hbar\omega)^2\right]\left[E_{CT}^2 - (2\hbar\omega)^2\right]}, \quad (1)$$

where $\hbar\omega$ is the incident photon energy, $q$ is the electronic charge, $E_{CT}$ is the CT state energy, $f_{CT}$ is its oscillator strength, and $\Delta\mu_{CT}$ is the difference in dipole moment between the ground and CT excited state. In comparison with $f_{CT} \sim 0.5$ and $\Delta\mu_{CT} \sim 7\,\mathrm{D}$ for a classic intramolecular CT NLO chromophore, such as 4-(N,N-dimethylamino)-4'-nitro-stilbene[26], intermolecular CT transitions at the DA interface of organic photovoltaic cells typically have a lower oscillator strength ($f_{CT} \sim 0.001 - 0.01$) and a larger change in dipole moment ($\Delta\mu_{CT} \sim 20\,\mathrm{D}$)[27].

In a DAS multilayer such as that shown in Fig. 1, net charge transfer and alignment of the CT transition is expected in the film normal direction ($\hat{z}$) owing to the planar $C_{\infty v}$ symmetry of the system. Consequently, when the DA hyperpolarizability tensor is dominated by the (scalar) intermolecular CT contribution, $\beta_{CT}$, along this direction, the second-order susceptibility tensor can be expressed as[13]:

$$\chi^{(2)}_{ijk} = F_{loc}\,\beta_{CT}\langle\Theta_{ijk}\rangle N_{CT}/t_{DAS}. \quad (2)$$

Here, $F_{loc}$ is an aggregate local field correction factor, $N_{CT}$ is the area density of interfacial CT states and $t_{DAS}$ is the DAS unit cell

thickness. The orientational order parameter, $\langle \Theta_{ijk} \rangle$, is averaged over all CT orientations defined by the angle, $\theta$, between their dipole moments and the $\hat{z}$ direction. Symmetry considerations dictate that there are only three unique, non-vanishing elements of the susceptibility tensor in this case, $\chi^{(2)}_{zzz}$, $\chi^{(2)}_{zii}$ and $\chi^{(2)}_{izi}$ (the index $i$ represents either of the in-plane dimensions, $x$ or $y$).

To isolate the various CT electronic and orientational factors that influence $\chi^{(2)}$, we begin by studying a single DA interface involving the donor pentacene (Pn) and the fullerene acceptor $C_{60}$. Bilayers with varying Pn and $C_{60}$ thicknesses are grown via vacuum thermal evaporation on sapphire substrates and characterized by measuring SHG with a weakly focused $\lambda_\omega = 800$ nm pump in the transmission geometry shown in Fig. 2a. Figure 2b verifies the quadratic scaling of the SH output

with pump intensity characteristic of all samples tested in this work; luminescence due to two photon absorption does not occur at the SH wavelength ($\lambda_{2\omega} = 400$ nm).

Figure 2c shows the SH power measured as a function of the sample tilt angle ($\theta$) for bilayers with 20 nm of $C_{60}$ deposited on top of Pn with varying thickness in the range 6 to 18 nm. As compared with either the bare $C_{60}$ or Pn reference layers, the Pn/$C_{60}$ bilayers yield an approximate six-fold increase in SH power that is independent of Pn thickness. Alternatively, fixing the underlying Pn thickness to 18 nm and varying the top $C_{60}$ thickness results in a similar effect, as shown in Fig. 2d. We note that there is no systematic variation in SH power with changing Pn or $C_{60}$ thickness in Fig. 2c,d, and thus the scatter in the data for different thicknesses should be interpreted mainly as the level

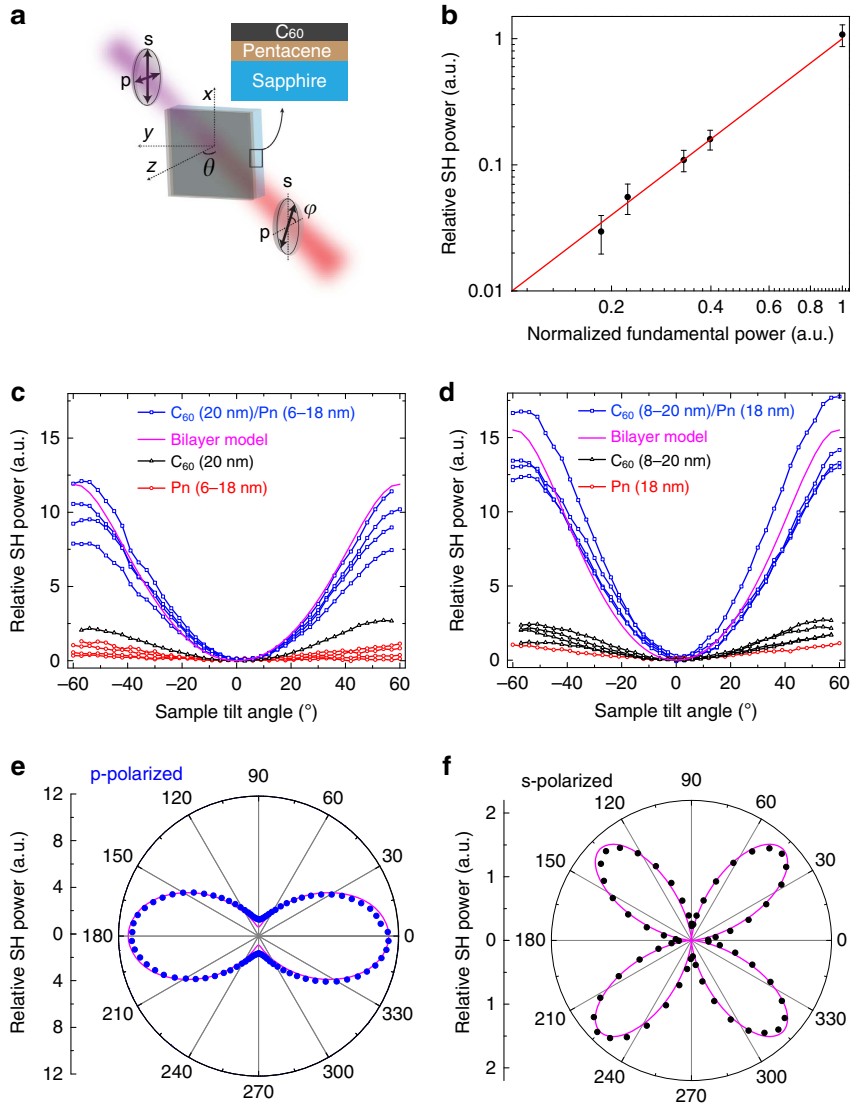

**Figure 2 | SHG from a single pentacene/C$_{60}$ interface.** (**a**) Experimental geometry used to measure transmitted second harmonic generation (SHG) as a function of the pump polarization ($\varphi$) and incidence ($\theta$) angles; the fundamental wavelength is $\lambda_\omega = 800$ nm. The inset shows the sample structure, which consists of a pentacene/C$_{60}$ bilayer deposited on a sapphire substrate. (**b**) Quadratic dependence (denoted by the red line) of SH power on fundamental power that is typical of all samples studied. The error bars reflect s.d.'s calculated from multiple measurements on the same sample structure. (**c**) Tilt scan results of p-polarized SHG from bilayers with fixed C$_{60}$ (20 nm) and varying pentacene thickness (blue squares) as indicated in the legend; the individual pentacene (red circles) and C$_{60}$ films (black triangles) are included for reference. (**d**) The same measurement carried out on bilayers with fixed pentacene (18 nm) and varying C$_{60}$ thickness. The strong increase in bilayer SHG independent of constituent film thickness points to a dominant contribution from the DA interface. (**e,f**) Polar plots show the dependence of p- and s-polarized SHG on pump polarization angle, consistent with a nonlinear susceptibility exhibiting $C_{\infty v}$ symmetry. Solid magenta lines in **c**–**f** are the result of global fits to all of the bilayer sample data associated with each respective panel using the surface SHG model described in Supplementary Note 1. The left-hand axes in **e** and **f** denote the radial scale.

of measurement-to-measurement and sample-to-sample uncertainty associated with measuring very weak SHG from a single interface. Taken together, these observations point to a strong nonlinear response associated with the Pn/$C_{60}$ interface that dominates over that associated with the organic film bulk, organic/air or organic/substrate interfaces.

Figure 2e and 2f, respectively, displays the dependence of p- and s-polarized SH power on the polarization angle of the fundamental beam ($\varphi$; Fig. 2a) obtained for a representative $C_{60}$ (12 nm)/Pn (18 nm) bilayer tilted at $\theta = 45°$. The data show that p-polarized SHG is in general much stronger than s-polarized SHG. Together with the observation that SHG maximizes at high tilt angle ($\theta$) in Fig. 2b,c, this suggests that the susceptibility tensor is dominated by $\chi_{zzz}^{(2)}$, which is qualitatively consistent with Pn/$C_{60}$ CT transitions oriented orthogonal to the DA interface.

The magnitude of the effective surface nonlinearity is determined by globally fitting the data in Fig. 2c–f with a simple model (Supplementary Note 1) calibrated against a z-cut LiNbO$_3$ reference wafer. The results, given in Voigt notation in terms of the effective surface NLO coefficients[28], are $d_{33}^s = \chi_{zzz}^{(2),s}/2 = 1.0 \times 10^{-13}$ e.s.u. ($420 \pm 76$ nm·pmV$^{-1}$), $d_{31}^s = \chi_{zii}^{(2),s}/2 = 2.5 \times 10^{-14}$ e.s.u. ($105 \pm 19$ nm·pmV$^{-1}$) and $d_{15}^s = \chi_{izi}^{(2),s}/2 = 2.5 \times 10^{-14}$ e.s.u. ($105 \pm 19$ nm·pmV$^{-1}$). The similarity of the latter two values suggests that Kleinman symmetry holds in this case, though this is not known a priori due to the proximity of the pump and CT transition energies. The value of $d_{33}^s$, which primarily results from the Pn/$C_{60}$ DA interface (Fig. 2c,d establish that contributions from other interfaces and the film bulk are small), is roughly two orders of magnitude larger than that typical for a monolayer of polar molecules adsorbed on a surface ($\sim 10^{-14} \sim 10^{-15}$ e.s.u.)[29]. Extrapolating this effective surface value to the bulk nonlinear coefficient that could be obtained in a DAS multilayer stack as in Fig. 1 and assuming a conservative estimate for the period ($t_{DAS} = 30$ nm in equation (2)) leads to $d_{33}^b = d_{33}^s/t_{DAS} \approx 14$ pmV$^{-1}$, which is similar in magnitude to that of the benchmark inorganic material LiNbO$_3$ ($d_{33} = 38$ pm V$^{-1}$ at 800 nm).

The ratio of tensor elements, $\chi_{zzz}^{(2),s}/\chi_{zii}^{(2),s} = 4.0 \pm 1.2$, found here is similar to that observed for poled polymer systems[30] and inorganic ABC nanolaminates[31–33]. Assuming a constant local field factor, this ratio depends only on the associated orientational averages, $\langle \Theta_{zzz} \rangle / \langle \Theta_{zii} \rangle = 2\langle \cos^3\theta \rangle / \langle \sin^2\theta \cos\theta \rangle$, and thus primarily reflects the width of the CT orientational distribution function, $f(\theta)$, since the average CT orientation can reasonably be expected to be normal to the interface (that is, $\langle \theta \rangle = 0$). Although $f(\theta)$ is not explicitly known, we assume it can be described by the general functional form $f(\theta) \propto \cos^n\theta$. In this case, $n \approx 2$ yields the observed $\langle \Theta_{zzz} \rangle / \langle \Theta_{zii} \rangle$ ratio determined above, from which it is subsequently possible to estimate the order parameter for CT state alignment[13], $S = \langle 3\cos^2\theta - 1 \rangle / 2 \approx 0.65$. This estimate agrees with the order parameter determined from dichroism in the linear CT absorption discussed below and is similar to the value observed for well-aligned nematic liquid crystals[34].

To quantify the CT electronic properties that underlie the hyperpolarizability in equation (1), CT absorption and EA measurements are carried out for the Pn/$C_{60}$ interface using the photovoltaic device architecture shown in Fig. 3a. Figure 3b plots the photovoltaic external quantum efficiency (EQE) measured for a typical device over several orders of magnitude on a logarithmic scale. The low-energy shoulder below the Pn and $C_{60}$ excitonic absorption manifold ($E < 1.4$ eV) is due to the CT transition at the DA interface. This shoulder is not observed for pure Pn or $C_{60}$ devices and its strong polarization anisotropy favouring p-polarized excitation is consistent with a transition dipole moment oriented normal to the layer interface.

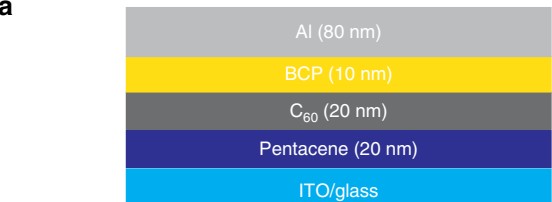

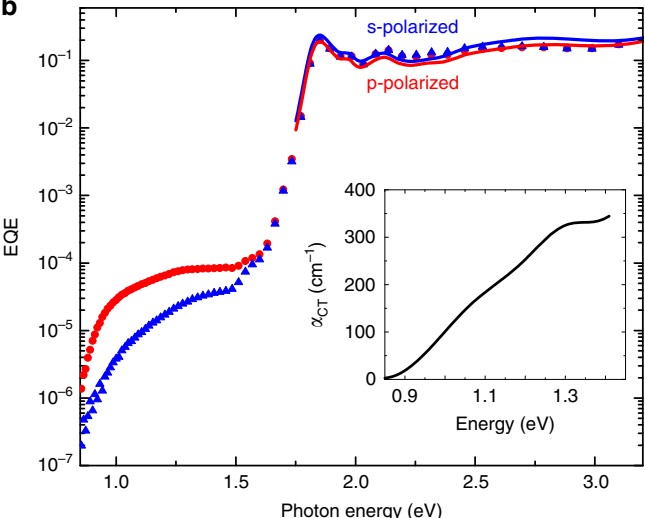

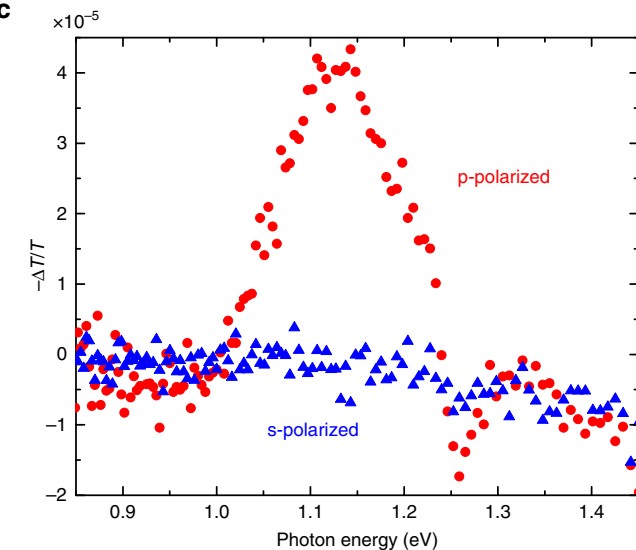

**Figure 3 | Absorption and electroabsorption spectra of pentacene/$C_{60}$ CT states.** (**a**) Schematic of the photovoltaic device structure used for external quantum efficiency (EQE) and electroabsorption (EA) measurements. (**b**) Polarized EQE spectra measured for the device. The s- and p-polarized EQE data overlap at energies >1.5 eV, where isotropic bulk layer exciton transitions dominate, but the polarizations differ markedly in the low-energy region ($E < 1.4$ eV) which is dominated by anisotropic CT-state absorption. The simulated s- and p-polarized EQE spectra (solid blue and red lines, respectively) in the excitonic region are based on the measured complex refractive indices of Pn and $C_{60}$ (Supplementary Fig. 1). The inset shows the CT absorption coefficient extracted from the EQE spectra. (**c**) Polarized EA spectra in the CT region ($E < 1.4$ eV), where the p-polarized response again dominates due to the alignment of CT states normal to the pentacene/$C_{60}$ interface.

Previous work has shown that the difference between p- and s-polarized EQE spectra provides a convenient route to isolating the anisotropic CT absorption from the largely isotropic, common mode excitonic background[35]. Coupled with transfer matrix modelling of the thin film optics using the known Pn and $C_{60}$ optical constants (Supplementary Fig. 1), a reasonable estimate of the CT absorption can be extracted from the EQE difference spectrum, $\Delta EQE = EQE_p - EQE_s$. The broad linewidth of the resulting CT spectrum may result from two, partially overlapping transitions[35]. Nevertheless, taking the molecular density at the DA interface ($\sim 10^{14}$ cm$^{-2}$) as an estimate for the density of available CT transitions, it is possible to integrate the resulting CT absorption cross-section as detailed in Supplementary Note 2 and determine an approximate value for the CT transition oscillator strength, $f_{CT} \sim 0.004$. In addition, it is also possible to estimate the dichroic ratio between normal ($\hat{z}$) and in-plane ($\|$) components of the CT absorption coefficient ellipsoid, $r \equiv \alpha_{CTz}/\alpha_{CT\|}$, and thereby independently determine[35] the nematic alignment order parameter, $S = (r-1)/(r+2) = 0.62$, which is consistent with that found from the nonlinear response above.

Strong polarization anisotropy is similarly observed in the CT spectral region ($E < 1.4$ eV) of the EA spectrum shown in Fig. 3c, confirming the highly oriented nature of the CT transition orthogonal to the DA interface. In this case, a dithered reverse bias is applied to the device to produce a $\hat{z}$-directed electric field, $F_z = F_{DC} + F_{AC} \cos(\omega t)$, and the change in absorption, $\Delta \alpha_{1\omega}$, is detected synchronously at the modulation frequency $\omega$. In contrast to the usual quadratic Stark effect observed in the excitonic region of the EA spectrum (not shown), the low-energy CT-related signal arises from the linear Stark effect owing to the net CT state alignment[36]. On the basis of the apparent change in oscillator strength observed for the lowest-energy CT transition, we estimate its associated dipole moment $\Delta \mu_{CT} \approx 40$ D (see Supplementary Note 3 for details). Taken together with the energy of this CT state (which is resolved from the EA peak at $E_{CT} = 1.12$ eV), its oscillator strength, and its degree of alignment determined above, these parameters provide a point of reference for comparison with established organic NLO-poled polymer systems.

**Sub-wavelength $\chi^{(2)}$ modulation.** Arguably the primary motivation for exploring intermolecular CT-based NLO materials lies in the opportunity to modulate $\chi^{(2)}$ at the nanoscale. To demonstrate this, we constructed asymmetric DAS multilayers via oblique-angle deposition on the surface of a shallow photoresist grating as illustrated in Fig. 4a. By uniformly depositing the D and S layers, and selectively depositing the A layers only on the 'windward' grating facets[37], it is possible to periodically vary the lateral distribution of DA interface and thus also the magnitude of $\chi^{(2)}$. In this case, rubrene, $C_{60}$ and the wide-gap molecule 4,4′-bis (N-carbazolyl)-1,1′-biphenyl (CBP) are chosen for the D, A and S layers, respectively, because they can be grown as amorphous thin, continuous layers to form a well-defined DAS stack. This is in contrast to DAS multilayer attempts with pentacene, which are complicated by strong crystallization that disrupts layer continuity (and thus CT state orientational order) within a few periods (see Supplementary Fig. 2 for details). The rubrene/$C_{60}$ interface nevertheless exhibits the same type of intermolecular CT-based $\chi^{(2)}$ nonlinearity established for pentacene/$C_{60}$, enabling bulk DAS multilayer stacks with $d_{33}^b \approx 9$ pm V$^{-1}$ (Supplementary Figs 3 and 4).

Figure 4b shows a cross-sectional scanning electron micrograph of an initial, bare sinusoidal photoresist grating on a Si substrate and Fig. 4c shows the result after deposition of a three

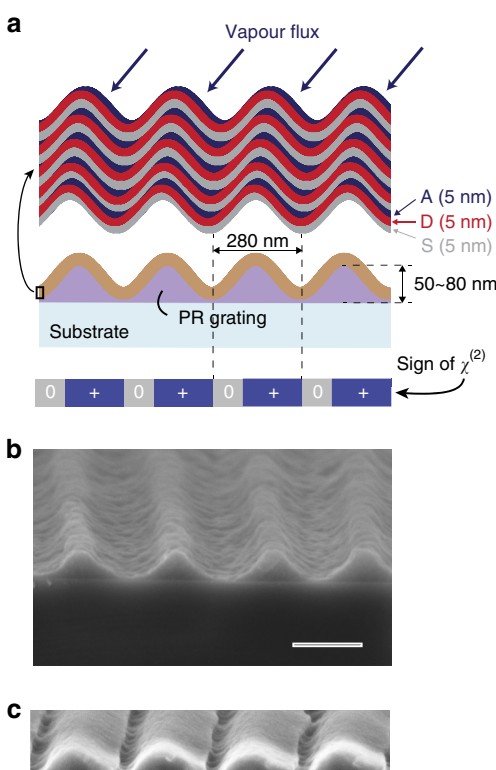

**Figure 4 | Sub-wavelength $\chi^{(2)}$ grating fabrication.** (**a**) Schematic showing the grating-shadowed oblique-angle deposition approach used to periodically pattern the distribution of donor–acceptor interface in a DAS multilayer. Samples consist of a sinusoidal photoresist grating on glass, with uniformly deposited CBP spacer and rubrene donor layers, and obliquely deposited $C_{60}$ acceptor layers. The nominal thickness of each layer is 5 nm and complete samples contain five DAS periods. (**b**,**c**) Cross-sectional scanning electron micrographs show the bare photoresist grating template (**b**) together with the result after coating with three DAS periods (**c**). The scale bar is 200 nm in both **b** and **c**. The individual layers in **c** are purposely grown with an exaggerated thickness of 25 nm for image clarity to enable the periodic accumulation of $C_{60}$ on the right-hand grating facets to be discerned.

period DAS multilayer using an oblique deposition angle of $\sim 70°$, with layer thicknesses exaggerated (individual D, A and S layers are each 25 nm thick) for image clarity. The periodic distribution of $C_{60}$ that results from the shadowing effect of the grating is clearly evident in the DAS example of Fig. 4c; actual SHG experiments were conducted using five DAS periods with 5 nm individual layer thickness to maximize the interface density as depicted in Fig. 4a.

Nonlinear diffraction measurements were used to verify the expected variation in $\chi^{(2)}$ according to the experimental geometry shown in Fig. 5a. Due to the short period of the grating, no diffracted orders are possible for the fundamental beam and only the $m = -1$ diffraction order is allowed for the SH according to the nonlinear grating equation, $\sin \theta_d = \sin \theta_i + m(\lambda_{2\omega}/\Lambda)$. Because the DA interface distribution in the oblique-angle deposited sample is spatially asymmetric with respect to the grating surface profile, these $\chi^{(2)}$-active regions will experience different local field intensities at the fundamental frequency,

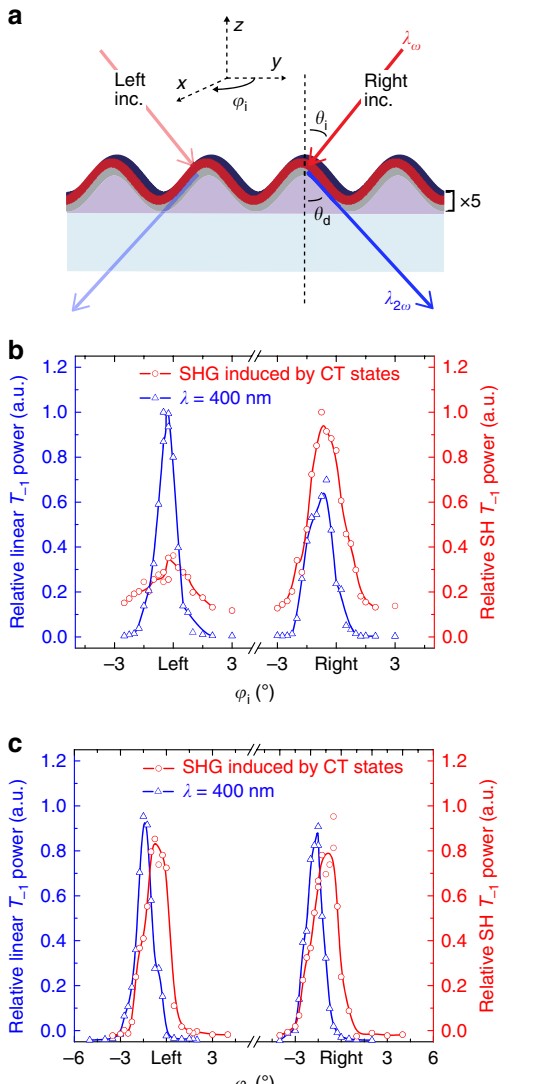

**Figure 5 | Second harmonic diffraction from $\chi^{(2)}$ gratings.** (**a**) Geometry of the nonlinear diffraction measurements. Only the zeroth and first diffracted orders are possible at the second harmonic (SH) owing to the $\Lambda = 280\,nm$ grating period. In the experiments, the incidence angle ($\theta_i$) of the p-polarized, $\lambda_\omega = 800\,nm$ fundamental beam is fixed and the intensity of the first transmitted SH order ($T_{-1}$) is monitored while rotating the sample azimuthally to exchange between left ($\varphi_i = 0°$) and right ($\varphi_i = 180°$) incidence on the grating. Linear diffraction is probed in the same manner using a low intensity, $\lambda = 400\,nm$ input beam generated by frequency doubling the pump with a beta barium borate (BBO) crystal. (**b**) Nonlinear SH and linear $T_{-1}$ diffraction intensities (both at $\lambda = 400\,nm$) measured for left and right incidence on the oblique-angle-modulated DAS sample. The SH diffraction is approximately three times stronger for right incidence, which cannot be explained by the linear $T_{-1}$ diffraction, since it instead favours left incidence and therefore a nonlinear $\chi^{(2)}$ grating must be present. (**c**) The same measurements conducted on a control grating with a uniformly deposited DAS multilayer are symmetric for left and right incidence.

when the pump beam is incident from the right or left. The magnitude of the resulting second-order nonlinear polarization grating is thus expected to differ for left and right incidence and, to a lesser extent, so is its diffraction efficiency since the coherently oscillating nonlinear polarization grating is also positioned asymmetrically with respect to the grating surface

profile. Collectively, these effects should lead to a strong asymmetry in diffracted SHG intensity for left versus right incidence if a $\chi^{(2)}$ grating exists.

Figure 5b shows the result of this experiment for the modulated DA sample in Fig. 4a based on the SH power detected in the $m = -1$ transmitted order ($T_{-1}$ detected at $\theta_d = -67.5°$) for a fundamental incidence angle, $\theta_i = 32.5°$. In this measurement, left and right incidence are exchanged simply by rotating the sample $\varphi_i = 180°$ about its azimuthal axis, $\hat{z}$, as shown in Fig. 5a. The data in Fig. 5b confirm a roughly three-fold difference in SH power for left versus right incidence that is not observed for a uniformly coated control grating in Fig. 5c. It is important to note, however, that the oblique-angle deposition process will also lead to some asymmetry in the linear index grating (analogous to a blaze due to shape asymmetry or due to the different $C_{60}$ refractive index on one grating facet), which will also contribute to the left/right asymmetry in diffracted SH power. To assess this possibility, linear diffraction measurements with low intensity light at the second harmonic wavelength (that is, $\lambda = 400\,nm$; blue lines in Fig. 5b,c) were conducted in the same manner. Slight asymmetry is indeed observed in the linear diffraction of the oblique-angle deposited sample in Fig. 5b (additional linear characterization is provided in Supplementary Fig. 5); however, it is opposite to that observed for the SH diffraction data and thus a nonlinear $\chi^{(2)}$ grating must exist in this sample to explain the SHG asymmetry.

These observations are understood on the basis of finite difference time domain (FDTD) nonlinear diffraction simulations in Fig. 6, which show the asymmetric optical field distribution ($|\mathbf{E}|^2$) of the grating together with the associated far-field SH diffraction intensities predicted for left versus right incidence at $\theta_i = 32.5°$. In Fig. 6a, when light at the fundamental frequency is incident from the right, the field intensity at the right-hand facets, where the DAS $\chi^{(2)}$ nonlinearity is located (denoted by the white dashed line) is ~2.5 times higher than when the fundamental is incident from the left. The induced second-order nonlinear polarization from the nonlinear regions is consequently much stronger for right incidence than for left incidence as shown in Fig. 6b, and this in turn leads to the asymmetry in $T_{-1}$ diffracted SH power summarized in Fig. 6c, in qualitative agreement with the experimental data.

## Discussion
While the $\chi^{(2)}$ nonlinearity of the DAS systems studied here compares favourably to that obtained recently for similar, inorganic ABC-type multilayers (where non-cancelling interfacial nonlinearities between three different oxide layers combine to enable $d_{33}^b \sim 0.2 - 6\,pm\,V^{-1}$)[31–33], it is important to point out that the values here likely involve a substantial resonant enhancement since the $\lambda_\omega = 800\,nm$ pump lies near the pentacene/$C_{60}$ and rubrene/$C_{60}$ CT transition energies ($E_{CT} = 1.12$ and $1.45\,eV$, respectively). Supplementary Fig. 4 quantifies the SHG resonance enhancement for rubrene/$C_{60}$. By the same measure, both donor materials and $C_{60}$ absorb strongly at the SH wavelength, which is not accounted for in our analysis and likely leads to an underestimation of the true SH response. This is, of course, problematic for SH or sum-frequency generation applications in the visible spectral range, but could be addressed by choosing donor and acceptor materials with wider excitonic energy gaps.

The ability to pair different donor and acceptor molecules also affords freedom to tune the CT state energy, which strongly impacts $\beta$ according to equation (1) and roughly follows the difference between donor ionization potential and acceptor electron affinity. Although low-CT oscillator strength seems like

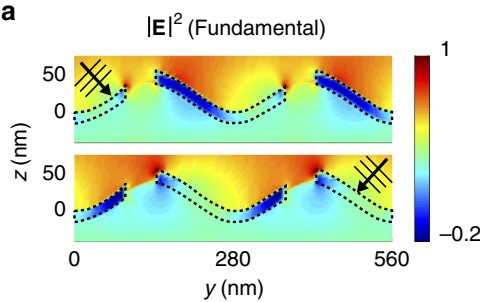

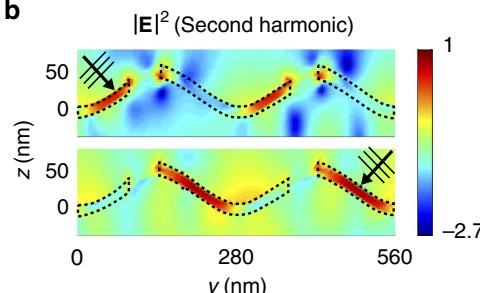

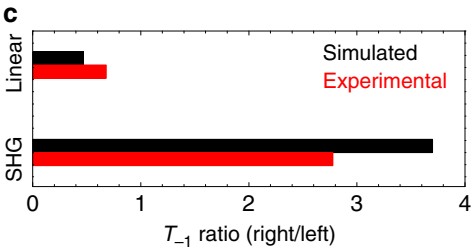

**Figure 6 | Numerical simulation of second harmonic diffraction. (a)** Finite difference time domain simulation of a nonlinear grating modelled after the oblique-angle-modulated DAS multilayer structure illustrated in Fig. 4a. The sinusoidal grating has a period $\Lambda = 280$ nm and a modulation depth $h = 50$ nm with a 15 nm thick nonlinear region defined on the right-hand grating facets indicated by dashed black lines. The top and bottom false-colour images show the relative optical field intensity ($|E|^2$, displayed with a logarithmic colour scale) at the fundamental frequency for left and right incidence, respectively, using the same $\theta_i = 32.5°$ as the experiments in Fig. 5. **(b)** Similar images showing the optical field intensity distribution produced at the second harmonic frequency. The asymmetry in second harmonic generation is a straightforward consequence of the larger fundamental field intensity in **a** that occurs in the nonlinear layer for right versus left incidence. **(c)** Ratios of the SHG and linear $T_{-1}$ diffraction intensities for right versus left incidence predicted from the model in comparison to the experimental results obtained from Fig. 5b.

an unavoidable trade-off for intermolecular CT states with small electron-hole wave function overlap, the large density of CT states per unit energy together with their potential for delocalization at a solid-state heterojunction may compensate to some degree[38]. In general, quantum chemical calculations are needed to predict optimal DA pairings and mutual orientations to maximize $\beta$, as well as to extend the design rules established for individual, intramolecular CT NLO chromophores to solid state, many molecule DA CT systems.

Apart from the electronic factors affecting $\beta$, CT orientational order at the DA interface (and thus $\chi^{(2)}$) can vary markedly depending on the morphology of the materials involved. Evidence suggests that the degree of CT state alignment tends to be higher at a smooth, largely crystalline DA interface than for an amorphous DA interface, presumably due to some degree of molecular-scale intermixing in the latter[35]. On the other hand, easily crystallizable molecules such as pentacene are problematic for DAS multilayers because they lead to a rapid increase in surface roughness after just a few periods (Supplementary Fig. 2), whereas amorphous materials (that is, rubrene in this case) enable smooth, many-period multilayers with little apparent degradation in interface quality, as evidenced by bulk-like (that is, quadratic) scaling of SHG with the number of DAS periods (see Supplementary Fig. 6 for details). Growth techniques such as organic molecular beam deposition provide one route to maintaining high interface quality/planarity in crystalline multilayers[39]. Other strategies such as sequential stamping or lamination of the different layers are also conceivable[40].

Beyond improving DAS $\chi^{(2)}$ materials, resonant waveguide grating schemes could be adopted to significantly increase the nonlinear conversion efficiency by choosing an appropriate combination of organic film thickness and modulation period[41]. In particular, our short-period $\chi^{(2)}$ patterning approach opens up an exciting possibility to quasi-phase match counterpropagating pump, signal and idler modes in an organic thin-film waveguide, potentially enabling mirrorless optical parametric oscillation[21]. Additional possibilities arise in two dimensions, where shadowed deposition enables a variety of complex nanoscale surface patterns[42] that could be harnessed to create new nonlinear metasurfaces.

In conclusion, we have explored the $\chi^{(2)}$ NLO response originating from intermolecular CT states at an organic semiconductor DA interface and have demonstrated the potential to achieve non-centrosymmetric multilayers with $d_{33} > 10$ pm V$^{-1}$. This interfacial approach to organic NLO materials introduces a general opportunity to structure the magnitude and sign of $\chi^{(2)}$ at the nanoscale, as demonstrated here by the fabrication of a $\Lambda = 280$ nm $\chi^{(2)}$ grating, which is the shortest reported to date. More broadly, given the semiconducting nature of the organic molecules involved, possibilities may also exist to dynamically modulate $\chi^{(2)}$ by perturbing the CT state with current or light stimuli, thus opening the door to new organic NLO materials whose nonlinear response can be controlled in both space and time.

## Methods

**Sample preparation.** Pentacene/C$_{60}$ bilayers used in SHG measurements are grown on pre-cleaned, ultraviolet–ozone-treated sapphire wafers in a thermal evaporator with a base pressure of $\sim 10^{-7}$ torr. Pentacene is deposited first at rates in the range 0.3–0.6 Å s$^{-1}$ with the substrate temperature held at 60 °C, followed by C$_{60}$ deposition at a rate of 0.5 Å s$^{-1}$ and substrate temperature of 20 °C. Pentacene/C$_{60}$ photovoltaic cells are fabricated in the same evaporator on pre-patterned indium tin oxide-coated glass substrates with the device structure: Pn (20 nm)/C$_{60}$ (20 nm)/bathocuproine (10 nm). The cells are completed by evaporating 80 nm Al contacts through a shadow mask to produce 4 mm$^2$ active device area. The Pn layer in these devices is intentionally grown at a rate of 1.5 Å s$^{-1}$ with a substrate temperature of $-25$ °C to minimize the grain size and achieve low shunt current to facilitate EA measurements. The different Pn grain size and morphology in the photovoltaic and SHG bilayers is a recognized uncertainty in extrapolating CT-state properties between the two cases. Gratings for the modulated $\chi^{(2)}$ samples are patterned in S1800 photoresist using inter-ference lithography with a $\lambda = 405$ nm write beam in a Llyod mirror configuration. Each DAS period is fabricated by thermally evaporating 4,4′-bis ($N$-carbazolyl)-1, 1′-biphenyl and rubrene layers with 5 nm nominal thickness at normal incidence, after which the sample is tilted to deposit 5 nm of C$_{60}$ at an oblique angle of 70°. This procedure is repeated to create a total of five DAS periods.

**Nonlinear characterization.** SHG measurements are performed using a Coherent Libra amplified Ti:sapphire pump laser ($\lambda = 800$ nm, 50 fs pulse width and 1 kHz repetition rate) weakly focused on the test samples with a beam waist of 2 mm and an approximate pulse energy of $\sim 40$ μJ; negligible SHG was detected from the sapphire substrate at this power level. The polarization of the fundamental beam is controlled using a half wave plate and samples are mounted on a rotation stage to vary the incidence angle. SH light is isolated from the fundamental through a series

of edge pass filters followed by two additional folding mirrors to reject residual scattered light. The SH signal is detected synchronously using a photomultiplier tube and a lock-in amplifier. To determine the absolute magnitude of the surface NLO $d$-coefficients, SHG is collected in direct transmission at the same pump power level from both the test samples and a congruently grown, $z$-cut $LiNbO_3$ reference crystal. The measured SH power of the thin-film test samples is fit as a function of the pump incidence and polarization angles to a nonlinear sheet polarization model for transmitted surface SHG, whereas the calibration SHG from $LiNbO_3$ is analysed via a thin-film model[43].

In the case of the $\chi^{(2)}$ grating measurements, the first order diffracted SHG power in transmission, $T_{-1}$, is measured as a function of the pump incidence angle, $\theta_i$, and the azimuthal sample rotation angle, $\varphi_i$, as illustrated in Fig. 5a. The test samples are aligned to maintain the pump focus on the same spot, irrespective of the two rotation angles to avoid error due to regional difference variations.

**Polarized EQE and EA.** Polarized EQE and EA measurements are carried out on Pn/$C_{60}$ photovoltaic devices using a laser-driven Xenon lamp filtered through a monochromator and chopped at 1 kHz. Polarization is controlled with a broadband wire grid polarizer. In EQE measurements, the light is incident onto the sample at an angle of ~45° and the photocurrent is pre-amplified and recorded by a lock-in amplifier. A total internal reflection waveguide geometry is used to enhance the weak CT EA signal (Supplementary Fig. 7), with incident light coupled into and out of the 1 mm thick substrate using a pair of right-angle prisms. A string of photovoltaic devices along the bounce path are electrically tied together and depleted by a $-4$ V reverse bias superimposed on a 2 Vrms, 1 kHz sinusoid. The relative change in transmitted light ($\Delta T/T$) is recorded with a photodiode and a lock-in amplifier. It is related to the field-induced change in absorption coefficient ($\Delta\alpha$) according to $N\Delta\alpha d = -\Delta T/T$, where $N \sim 12$ is the total number of reflections along the waveguide. Neglecting interference effects, the optical interaction length, $d$, through each device is taken to be approximately twice the Pn or $C_{60}$ layer thicknesses divided by the cosine of the incidence angle (due to the double pass in reflection off the cathode mirror) for interpretation of their respective (bulk excitonic) EA signals, whereas 1 nm is assumed for the effective thickness of the interfacial CT region.

**Nonlinear FDTD simulations.** The near and far fields of both fundamental ($\lambda = 800$ nm) and SH ($\lambda = 400$ nm) waves are calculated using the nonlinear material implementation of the commercial FDTD software package, Lumerical Solutions. The grating geometry and period are taken according to the cross-sectional images in Fig. 4, with $\Lambda = 280$ nm, $h = 50$ nm and a non-dispersive refractive index $n = 1.6$ taken for the photoresist. Nonlinear thin films (15 nm thick, linear index $n = 2$) are inserted on the right-hand facets and cover ~80% of the grating period to mimic the DA interface distribution, with a single nonlinear susceptibility element, $\chi^{(2)}_{zzz}$, set to 40 pm V$^{-1}$. The pump is taken to be a plane wave incident from the left or right at 32.5° with respect to the sample normal. Bloch boundary conditions are implemented in the $\hat{y}$ direction, whereas perfectly matched layers are used to model open boundary conditions in the $\hat{x}$ and $\hat{z}$ directions. Far-field diffraction intensities are calculated from the electromagnetic near fields on a virtual surface using Green's theorem[44].

**Data availability.** The data that support the findings of this study are available from the corresponding author on reasonable request.

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

## Acknowledgements

We thank Dr Bangzhi Liu for helping with scanning electron microscopy images. This work was supported in part by the Air Force Office of Scientific Research Young Investigator Program under Award FA-9550-14-1-0301 and by the the U.S. Department of Energy, Office of Basic Energy Sciences under Award DE-SC0012365. Ya.Y. and V.G. were supported by the U.S. Department of Energy under Award DE-SC0012375.

## Author contributions

Yi.Y. designed and performed the experiments, derived the models and analysed the data. Ya.Y. assisted with SHG measurements and B.W. helped carry out the FDTD simulations. Yi.Y and N.C.G wrote the manuscript in consultation with B.W., Ya.Y. and V.G.

## Additional information

**Competing financial interests**: The authors declare no competing financial interests.

**How to cite this article**: Yan, Y. *et al.* Sub-wavelength modulation of $\chi^{(2)}$ optical nonlinearity in organic thin films. *Nat. Commun.* **8**, 14269 doi: 10.1038/ncomms14269 (2017).

**Publisher's note**: 

