## [Peer Review File · Nature Communications]

Reviewers' comments:

Reviewer #1 (Remarks to the Author):

This manuscript reports on a relatively new scheme for nonlinear optical frequency conversion, based on charge transfer through donor-acceptor interfaces, as opposed to the more classical donor-acceptor conjugated molecule template. Theory based proposition of this mechanism had been reported by Santo Di Bella et al. some 20 years ago, followed by actual demonstration by Kajzar et al. in 1998 and 2001, with the demonstration of both second and third harmonic generation enhancement. One of the layer is made of the same C60 moiety as in this manuscript. Short of a conceptual advance, this manuscript is showing progress on that same direction as compared to earlier work in terms of methodology and fabrication technology. As for the efficiency, a factor of 2 (from a little below 5 pm/V to 10 pm/V) cannot be considered as a decisive advance. Methodology wise, the authors are deploying an impeccable range of measurement techniques that is derived from their solid knowledge of electronic and photovoltaic processes in organic blends and superstructures. Complementary to the central NLO experiments, their use of photovoltaic EQE and electro-absorption spectra brings considerable reinforcement to the otherwise scarce NLO figures, as well as helping to reinforce the interpretation.

The fabrication methods, their use of shadowed oblique angle deposition on top of a pre-patterned grating of relatively short period (280 nm), is a central feature of the paper, as it provides both alternation of the interfaces from active to passive (akin to a class of periodically poled nonlinear waveguide structure) and adequate structure for nonlinear diffraction by the $m=1$ or -1 orders. Demonstration of the angular asymmetry of the diffraction is consistent with the reported structure. This fabrication technique had been proposed earlier and used in another context (that of PT symmetry in organic structures) by the authors.

This manuscript is indeed impressive in terms of its methodological rigor and thoroughness, combining a solid know-how in both photovoltaics and NLO, which makes for an unusual combination that is bound to produce more original outputs in the future, which are alluded to in the conclusion. It also embodies a fabrication tour de force, based on a clever deposition technique allowing for a desirable alternation of structures.

However, although the result therein, namely reasonably efficient NLO diffraction, can be considered new as compared to earlier effects by the same technique, none of the individual building blocks leading to the final result are original as such. Clever assembly of such steps could have made for an overall added-value, be in terms of originality of the effect, remarkable efficiency of the targeted effect if not original, interesting demonstrator or application, but none of these criteria is met here.

Indeed, second-harmonic generation, be it in propagative, surface, diffracted or wave-guided mode can be found for decades in a wealth of publications. The diffraction mode chosen here is anything but new and in some ways, unless an original device application is demonstrated which is not the case here, cannot be considered superior in terms of applications to say wave-guiding, which is not considered here (see for comparison the electro-optic diffraction from a NLO grating, Hybrid semiconductor polymer resonant grating waveguide structures, Levy Yurista et al., Optical Materials 17, 149 (2001)

which demonstrates electrically driven modulation and extremely narrow line-width). The reported NLO coefficient of the order of 10 pm/V is satisfactory, but not impressive as compared to 30 pm/V for ferroelectric Lithium Niobate and more for other organic as well as inorganic NLO materials.

Moreover, the short grating period (280 nm) is presented as a major achievement, whereas it is commensurable to half the harmonic wavelength, which entails no new physics as is within the diffraction limit.

Other topical comments or possibly negative aspects:

The two level-model (equation 1) should be referred to the original citation.

Page 5: it seems strange that the quadratic S order parameter ($\sim \cos 2\theta$) can be inferred from the cubic ($\sim \cos 3\theta$) order parameter. Moreover, the S parameter is wrongly termed by the authors under "polar order parameter" (which would then be $\sim \cos \theta$ whereas it is in fact an axial or more

precisely a quadrupolar parameter. The authors should clarify their approach.

Page 8, line 191, the nonlinear diffraction experiment should read:

$n(2\omega) \cdot (\sin\theta)^i = n(\omega) \cdot (\sin\theta)^d + m(\lambda(2\omega))$;)

The index of refraction, or effective indices, cannot be omitted, as in the manuscript, or ascribed to the wavelengths. The authors should clarify this and detail the value of the indices at both fundamental and harmonic frequencies.

In conclusion, this remarkable work should be submitted to a first rate topical journal in optics and material science, such as ACS Photonics or Advanced Optical Materials, among other possibilities, but fails to meet the criteria for consideration in Nature, due to originality issues, as well as the lack of a clear-cut conceptual or applicative breakthrough.

Reviewer #2 (Remarks to the Author):

I found this an interesting study, very much presented in two parts. Demonstrating materials suitable for interfacial CT states and subsequent SHG, and a novel approach (and demonstration) of nanoscale patterning.

The use of artificial (non-centrosymmetric) structures to enhance NLO has been established for some years. The authors themselves fully acknowledge this and provide some useful references.

The initial bilayer work, on a single pentacene-C60 DA interface (Fig.2), verifies a strong nonlinear response is detected and confirms a significant DA dipole orientation across an interface (aligned orthogonal to the interface by the stronger p-pol response). Further studies using EA and EQE spectra, supported by a transfer matrix approach, provide a reasonable estimate of the CT properties and useful parameters (oscillator strength, CT energy, etc.), all allowing some degree of comparison with other NLO organic systems. The experiments themselves are a rather nice way of probing CT states and are only recently disclosed (ref. 32).

The subsequent work explores modulating the intermolecular CT-based NLO materials on the nanoscale – and in my opinion, perhaps better represents the novelty and ideas of the work. The titled shadowing technique is itself fairly well known – but, as far as I am aware, I haven't come across it applied as the present work does. It is a neat approach that, as the authors show, has been successful.

The key results and discussions around Figure 5a (and 5b for reference) clearly show the realization of nanoscale patterning of chi (2). The observed asymmetry in diffracted SHG intensity for left vs. right incidence show this very well. The authors have also carefully thought about ruling out the asymmetry effects due to 'blaze' of the grating – although, as noted below, I would've expected/hoped to see some basic (transmission) spectroscopy of the structure; useful for verifying the degree of asymmetry. Particularly given the authors own discussions highlighting “..that both donor materials and C60 absorb strongly at the second harmonic wavelength” which is “..not accounted for in our analysis and likely leads to an underestimation of the true SH response”. Interested readers may like to investigate and further develop related work themselves.

I'm not wholly convinced with SEM images (only as a supporting role) – nor I hasten to add, have I much expertise with them, but personally I wonder whether some basic spectroscopy (transmission) on the grating samples could have also been included (e.g. in the SI). The main SHG emission results show some very convincing radiation profiles, so rotation stage set-ups are presumably also available for basic white-light transmission spectroscopy of the grating structure. Such spectra would also probe the degree of grating asymmetry (the blaze) from a more conventional perspective – likely to be more familiar to a wider audience.

The authors have carried out some FDTD nonlinear diffraction simulations and have been used well to support the observations. Upon reading, however, I felt given the considerable work that has gone into these perhaps more analysis could've been shown or expanded on. For example, in Figure 6 the fields (for the fundamental and SHG frequencies) in the two cases are not quantitatively compared (only by false colours) – what is the scaling of the fields between 6a and 6b?

Given that the results show very clear qualitative agreement with the observations, I was hoping the authors might have some discussion (and examples) about trying to further develop the nanoscale patterning of $\chi^{(2)}$ concept. During the discussions, the authors noted several further opportunities concerning different materials but not so much on the nanoscale patterning – what was the significance of choosing $\Lambda=280$ nm? It needn't be an exhaustive study but, e.g. can the choice of the grating period help enhance the SHG conversion? In this regard can the grating dimensions also be used as a resonator? I view the patterning (and the demonstration of it) as an important part of the study, adding to the general area of using artificial (non-centrosymmetric) structures to enhance NLO.

Overall, I believe it is the 'sum-of-the-parts' here that would make the report suitable for the wider readership of Nature Communications. Individually the parts demonstrate some good experimental techniques, good quality NLO characterisation and interesting proof of concept of the nanoscale patterning. I think addressing some of the comments above, particularly highlighting the further developments and opportunities available the work will rightly attract some considerable interest.

Reviewer #3 (Remarks to the Author):

The authors demonstrate that intermolecular charge transfer states between separate donor and acceptor molecules can be used to achieve high $\chi^{(2)}$ films. The authors performed polarized SHG to measure the NLO coefficients. Furthermore they fabricate a grating of periodicity 280 nm and show that it acts as a grating for SHG. The authors have performed fundamental studies of SHG in thin modulated films and I recommend that this paper be published after they clarify and correct a number of points

a) They describe the need to periodically modulate $\chi^{(2)}$ for quasi phase matching (QPM); Figure 1b and 3rd paragraph. However they do not give the equation for QPM coherence length. It depends inversely on the difference in refractive indices at fundamental and SHG wavelengths. It would be useful for them to calculate coherence length as a function of wavelength.

b) It is not clear why they chose 280 nm as modulation length. Was that the coherence length at 800 nm fundamental wavelength for their material system pentacene/rubrene with C60?

c) The $\chi^{(2)}$ that they report may be resonantly enhanced since SHG occurs at 400 nm. It would be useful to comment on the value of $\chi^{(2)}$ away from 400 nm. How does it compare with LiNbO₃? They also say in Discussion section paragraph 1 "both donor materials and C60 absorb strongly at the SHG wavelength, which is not accounted for in our analysis". This is a major issue with use of organics in the visible wavelength. It would be more useful to study organics for parametric generation in the near IR away from absorption wavelengths

d) the authors should reference the first paper on QPM in solid organic/polymer films: Khanarian, G., Norwood, R.A., Haas, D., Feuer, B. and Karim, D., 1990. Phase-matched second-harmonic generation in a polymer waveguide. Applied Physics Letters, 57(10), pp.977-979.

e) "ABC -type multilayers" in Discussion paragraph 1, please clarify what this means

We appreciate the informative feedback and constructive criticism from all of the reviewers. We have addressed each comment/question by the reviewers as detailed below. Original reviewer comments are listed in italics, our responses are in black, and revisions in the manuscript are highlighted in yellow.

Response to reviewer # 1:

As for the efficiency, a factor of 2 (from a little below 5 pm/V to 10 pm/V) cannot be considered as a decisive advance.

...although the result therein, namely reasonably efficient NLO diffraction, can be considered new as compared to earlier effects by the same technique, none of the individual building blocks leading to the final result are original as such. Clever assembly of such steps could have made for an overall added-value, be in terms of originality of the effect, remarkable efficiency of the targeted effect if not original, interesting demonstrator or application, but none of these criteria is met here.

We emphasize that the point of our manuscript is not to best the nonlinear performance of existing materials or the previous work by Kajzar et. al., but rather to introduce a new opportunity for spatially structuring $\chi^{(2)}$ at small length scales; our title is intended to emphasize this. While we don't dispute that the individual building blocks that enable this result (i.e. $\chi^{(2)}$ response from intermolecular CT states and patterning by shadowed oblique angle deposition) are well established, the recognition that these two concepts can be combined to structure $\chi^{(2)}$ at the nanoscale is novel and has not previously been reported. This advance is simple in nature but nevertheless carries high significance because it changes sub-wavelength patterning of $\chi^{(2)}$ from a heroic effort into something that is routinely possible in many labs, and it lays the groundwork for truly nanostructuring $\chi^{(2)}$ (e.g. on the 10-100 nm length scale), which is not currently possible with any other technique.

...the short grating period (280 nm) is presented as a major achievement, whereas it is commensurable to half the harmonic wavelength, which entails no new physics as is within the diffraction limit.

Our choice of the 280 nm grating period was driven only by the need to clearly demonstrate that we have in fact modulated $\chi^{(2)}$, and asymmetric SHG diffraction measurements were the simplest unequivocal experiment for us to perform. The chosen period results from a combination of the minimum that we can reliably write with our $\lambda=405$ nm interference lithography system and the need to retain at least one propagating second harmonic diffracted order to measure - which is only possible for periods exceeding

267 nm based on our 800 nm fundamental wavelength (i.e. $\Lambda_{\min} = [\lambda_{2\omega}^{-1} + \lambda_{1\omega}^{-1}]^{-1}$ to ensure a propagating second harmonic diffracted order). Nevertheless, it should be self-evident based on this demonstration that the period could be far smaller since the shadowed oblique angle deposition approach works for arbitrarily short periods.

The two level-model (equation 1) should be referred to the original citation.

We have added the original references on the two level model formulation by Oudar & Chemla [*J. Phys. Chem.* **66** 2664 (1977) and *J. Phys. Chem.* **67** 446 (1977)] to our reference list (now Refs 24 and 25). The paragraph at the bottom of page 2 of the manuscript now reads:

"The hyperpolarizability associated with a given CT transition can be qualitatively understood from a simple two-level model^{24,25}, which predicts: ..."

Page 5: it seems strange that the quadratic S order parameter ($\sim \cos^2\theta$) can be inferred from the cubic ($\sim \cos^3\theta$) order parameter. Moreover, the S parameter is wrongly termed by the authors under "polar order parameter" (which would then be $\sim \cos\theta$ whereas it is in fact an axial or more precisely a quadrupolar parameter. The authors should clarify their approach.

We did not directly deduce the order parameter, S , from $\langle \cos^3 \theta \rangle$. Rather we proceeded by assuming a reasonable form for the orientation distribution function, $f(\theta) = \cos^n \theta$ (typically found, e.g. for nematic liquid crystals) and then determining what value the exponent n must take in order to reproduce the $\chi^{(2)}$ tensor element ratio $\langle \Theta_{zzz} \rangle / \langle \Theta_{zii} \rangle = 2 \langle \cos^3 \theta \rangle / \langle \sin^2 \theta \cos \theta \rangle = 4$ that we observe from our SHG measurements in Fig. 2. This ratio simplifies to $\langle \cos^3 \theta \rangle / \langle \cos \theta \rangle = 2/3$ and, by inserting the assumed form of the distribution function, $f(\theta) = \cos^n \theta$, it is straightforward to calculate $n \approx 2$, from which we subsequently evaluate $S = \frac{1}{2} \langle 3 \cos^2 \theta - 1 \rangle = 0.65$. We acknowledge that this is not a rigorous determination of S , however, it provides a good first approximation and agrees well with our subsequent, independent determination of S based on CT absorption dichroism later in the manuscript in Fig 3.

We have eliminated use of the term 'polar' in the manuscript and have attempted to clarify our estimation procedure for the order parameter by revising the paragraph on page 5 of the manuscript to read:

Although $f(\theta)$ is not explicitly known, we assume it can be reasonably described by the general functional form, $f(\theta) \propto \cos^n \theta$. In this case, $n \approx 2$ yields the observed $\langle \Theta_{zzz} \rangle / \langle \Theta_{zii} \rangle$ ratio determined above, from which it is subsequently possible to estimate the order parameter for CT state alignment, $S = \langle 3 \cos^2 \theta - 1 \rangle / 2 \approx 0.65$.

*Page 8, line 191, the nonlinear diffraction experiment should read:
 $n(2\omega). (\sin\theta)^i = n(\omega). (\sin\theta)^{d+m} (\lambda(2\omega)/\Lambda)$*

The index of refraction, or effective indices, cannot be omitted, as in the manuscript, or ascribed to the wavelengths. The authors should clarify this and detail the value of the indices at both fundamental and harmonic frequencies.

Because all of the incident and diffracted orders that we observe are propagating in air (as opposed to coupling into a guided mode of the film), the relevant fundamental and second harmonic indices are both that of air and thus can safely be dropped from the equation.

Responses to reviewer #2:

I would've expected/hoped to see some basic (transmission) spectroscopy of the structure; useful for verifying the degree of asymmetry... personally I wonder whether some basic spectroscopy (transmission) on the grating samples could have also been included (e.g. in the SI). The main SHG emission results show some very convincing radiation profiles, so rotation stage set-ups are presumably also available for basic white-light transmission spectroscopy of the grating structure. Such spectra would also probe the degree of grating asymmetry (the blaze) from a more conventional perspective – likely to be more familiar to a wider audience.

Per the reviewer's suggestion, we have conducted additional experiments to more completely characterize the linear diffraction asymmetries of the nonlinear grating samples in Fig. 5. Figure 1 below displays simple white light transmission spectra (i.e. the zeroth diffracted order, T_0) for incidence angles $\theta_i = 35^\circ$ and $\theta_i = -35^\circ$ (i.e. close to that used for the 'left' and 'right' nonlinear diffraction measurements in Fig. 5) of the oblique-angle deposited DAS multilayers. The similar spectra indicate a negligible blazing effect for T_0 at both the fundamental and second harmonic wavelengths.

Figure 1. Linear transmission spectra of the oblique angle deposited grating samples from Fig. 5b in the text. The data are obtained near the same incidence angles that the nonlinear diffraction measurements are taken.

In Fig. 2 below, we have measured the absolute -1 order transmitted linear diffraction efficiency (that is, T_{-1}) for a $\lambda = 405$ nm laser beam close to the second harmonic wavelength as a function of 'left' and 'right' incidence angle; it is not possible to measure the diffraction efficiency of the fundamental since the short grating period does not allow any propagating orders. The slight asymmetry favoring diffraction from left incidence angles is consistent with that observed in Fig. 5b in the main text and confirms a slight blazing effect in the linear T_{-1} diffracted order due to the oblique angle deposition of C_{60} . This effect is qualitatively captured in the numerical grating simulations in Fig. 6 of the main text and stems from the refractive index contrast between C_{60} and the surrounding organic materials at $\lambda \sim 400$ nm (see, for example, Fig. S2).

Figure 2. Linear diffraction efficiency at $\lambda = 405$ nm for the first transmitted order (T_{-1}) of the oblique angle deposited gratings used in the nonlinear measurements of Fig. 5b in the main text.

We have added these two plots into the revised Supplementary Information as Fig. S7 and changed the main text at the bottom of page 9 in the manuscript to read:

"Slight asymmetry is indeed observed in the linear diffraction of the oblique angle-deposited sample in Fig. 5b (additional linear characterization is provided in Supplementary Fig. S7); however, it is opposite..."

The authors have carried out some FDTD nonlinear diffraction simulations and have been used well to support the observations. Upon reading, however, I felt given the considerable work that has gone into these perhaps more analysis could've been shown or expanded on. For example, in Figure 6 the fields (for the fundamental and SHG frequencies) in the two cases are not quantitatively compared (only by false colours) – what is the scaling of the fields between 6a and 6b?

Our omission of numbered scale bars here was an error; these have been added in to the revised Fig. 6 shown below.

In addition, we have changed the caption for Fig. 6 to read:

"The top and bottom false color images show the **relative** optical field intensity ($|E|^2$, **displayed with a logarithmic color scale**) at the fundamental frequency for left and right incidence, respectively..."

Given that the results show very clear qualitative agreement with the observations, I was hoping the authors might have some discussion (and examples) about trying to further develop the nanoscale patterning of $\chi^{(2)}$ concept. During the discussions, the authors noted several further opportunities concerning different materials but not so much on the nanoscale patterning – what was the significance of choosing $\Lambda=280$ nm? It needn't be an exhaustive study but, e.g. can the choice of the grating period help enhance the SHG conversion? In this regard can the grating dimensions also be used as a resonator? I view the patterning (and the demonstration of it) as an important part of the study, adding to the general area of using artificial (non-centrosymmetric) structures to enhance NLO.

Our choice of the 280 nm grating period was driven by the need to clearly demonstrate that we have in fact modulated $\chi^{(2)}$, and asymmetric SHG diffraction measurements were the simplest unequivocal experiment for us to perform. The chosen period results from a combination of the minimum that we can reliably write with our $\lambda=405$ nm interference lithography system, and the need to retain at least one propagating second harmonic diffracted order to measure - which is only possible for periods exceeding 267 nm based on our 800 nm fundamental wavelength (i.e. $\Lambda_{\min} = [\lambda_{2\omega}^{-1} + \lambda_{1\omega}^{-1}]^{-1}$ to ensure a propagating second harmonic diffracted order).

In regard to patterning architectures for enhancing SHG conversion, the simplest route is to adopt a resonant waveguide enhanced structure by increasing the organic film thickness to support a guided mode at the fundamental wavelength for increased nonlinear interaction. This approach has previously demonstrated >1000-fold improvement in SHG conversion efficiency [see for example, A. Saari *et. al. Opt. Exp.* **18**, 12298 (2010) among many other references]. More generally, though, short period $\chi^{(2)}$ modulation opens up new parametric processes, as illustrated in Fig. 3 below. By choosing the thickness of the organic film and the modulation period appropriately, it is possible to phase match between pump, signal, and idler modes to achieve mirrorless optical parametric oscillation.

Figure 3. Overview of mirrorless optical parametric oscillation enabled by short-period $\chi^{(2)}$ modulation in an organic thin film waveguide.

Figure 3b details this possibility further, showing the energy and momentum-conserving transitions in the dispersion diagram calculated for an asymmetrically-clad organic thin film waveguide indicated in the inset. In this case, we expect transverse magnetic (TM) modes to couple most strongly to the normally-oriented CT nonlinearity in the film and have thus drawn the transitions accordingly. By employing a $\chi^{(2)}$ modulation period (e.g. grating periodicity) of order ~ 600 nm, sufficient momentum is enabled (black arrow) to quasi-phase match the parametric conversion into signal and counterpropagating idler waves in the TM_1 and TM_0 modes of the film at energies of 1.15 eV and 0.5 eV, respectively. At an idler wavelength of nearly $2.5 \mu\text{m}$, an infrared transparent substrate such as MgF_2 or sapphire would be necessary, however, the organic film itself is expected to remain transparent in this spectral region for wavelengths up to $\sim 4 \mu\text{m}$ based on the known infrared absorption of organic semiconductors such as C_{60} .

Perhaps the most intriguing opportunity we are pursuing with this $\chi^{(2)}$ patterning approach is the creation of nonlinear metasurfaces. Nemiroski and co-workers (ACS Nano, 8, 11061 (2014)) have shown a wide variety of complex 2D patterns by tailoring oblique angle deposition conditions (see Fig. 4 below for an example), which in turn is allowing us to achieve nonlinear metasurfaces with complex $\chi^{(2)}$ variation for a number of different applications.

To address the patterning opportunities that this work opens up, we have added the following paragraph to the end of the discussion section on page 11:

"Beyond improving DAS $\chi^{(2)}$ materials, resonant waveguide grating schemes could be adopted to significantly increase the nonlinear conversion efficiency by choosing an appropriate combination of organic film thickness and modulation period⁴¹. In particular, our short-period $\chi^{(2)}$ patterning approach opens up the opportunity to quasi-phase match counterpropagating pump, signal, and idler modes in an organic thin film waveguide, potentially leading to mirrorless optical parametric oscillation²¹. Additional possibilities arise in two dimensions, where shadowed deposition enables a variety of complex nanoscale surface patterns⁴² that could be harnessed to create new nonlinear metasurfaces."

Figure 4. Overview of several different 2D patterns that can be accomplished with oblique angle deposition conditions. Figure taken from Nemiroski et. al., ACS Nano, 8, 11061 (2014).

Responses to reviewer #3:

- a) *They describe the need to periodically modulate $X(2)$ for quasi phase matching (QPM); Figure 1b and 3rd paragraph. However they do not give the equation for QPM coherence length. It depends inversely on the difference in refractive indices at fundamental and SHG wavelengths. It would be useful for them to calculate coherence length as a function of wavelength.*

Based on ellipsometry measurements, the complex refractive indices that effectively characterize our pentacene/C₆₀ composite films are $\tilde{n}_{400} = 2.04 + 0.13i$ and $\tilde{n}_{800} = 1.90 + 0.00i$, at the second harmonic and fundamental wavelengths, respectively. If we were considering co-propagating modes in this medium, the QPM coherence length is $l_{coh} = \frac{2\pi}{2k_\omega - k_{2\omega}} = \frac{\lambda_\omega}{2(n_{2\omega} - n_\omega)} \approx 2.8 \mu\text{m}$, which could be achieved with existing poling techniques. However, to quasi-phase match contrapropagating modes (i.e. backward phase matching as emphasized in the manuscript), the QPM coherence length is instead $l_{coh} = \frac{2\pi}{2k_\omega + k_{2\omega}} = \frac{\lambda_\omega}{2(n_{2\omega} + n_\omega)} \approx 100 \text{ nm}$, which is far below what is possible with previously existing $\chi^{(2)}$ patterning techniques.

- b) *It is not clear why they chose 280 nm as modulation length. Was that the coherence length at 800 nm fundamental wavelength for their material system pentacene/rubrene with C60?*

Our choice of the 280 nm grating period was driven by the need to clearly demonstrate that we have in fact modulated $\chi^{(2)}$; asymmetric SHG diffraction measurements were the simplest unequivocal experiment for us to perform. The chosen period results from a combination of the minimum that we can reliably write with our $\lambda = 405 \text{ nm}$ interference lithography system, and the need to retain at least one propagating second harmonic diffracted order to measure in the experiment - which is only possible for periods exceeding 267 nm based on our 800 nm fundamental wavelength (i.e. $\Lambda_{\min} = [\lambda_{2\omega}^{-1} + \lambda_{1\omega}^{-1}]^{-1}$ to ensure a propagating second harmonic diffracted order).

- c) *The $X(2)$ that they report may be resonantly enhanced since SHG occurs at 400 nm. It would be useful to comment on the value of $X(2)$ away from 400 nm. How does it compare with LiNbO₃? They also say in Discussion section paragraph 1 "both donor materials and C60 absorb strongly at the SHG wavelength, which is not accounted for in our analysis". This is a major issue with use of organics in the visible wavelength. It would be more useful to study organics for parametric generation in the near IR away from absorption wavelengths.*

Supplementary Figure S6b directly quantifies the degree of resonant enhancement by showing the SHG excitation spectrum for the rubrene/C₆₀ material system. From this plot, the non-resonant SHG output power (as taken from the measurement at 1000 nm fundamental wavelength) is roughly a factor of 3.5 lower than that for 800 nm fundamental

wavelength (400 nm SHG), which corresponds to a factor of $\sqrt{3.5} \approx 1.9$ x resonance enhancement in $\chi^{(2)}$ at a fundamental wavelength of 800 nm.

As noted in the second paragraph on page 10 of the manuscript, we recognize the problematic nature of second harmonic (or sum frequency) generation in the presence of strong organic absorption in the visible for these particular material systems, but point out that different donor and acceptor molecules with wider excitonic energy gaps could be chosen to circumvent this problem. Nevertheless, we are in full agreement regarding opportunities to study parametric processes in the near infrared; this is a focus of our current work.

d) the authors should reference the first paper on QPM in solid organic/polymer films: Khanarian, G., Norwood, R.A., Haas, D., Feuer, B. and Karim, D., 1990. Phase - matched second - harmonic generation in a polymer waveguide. Applied Physics Letters, 57(10), pp.977-979.

We have added this citation as reference 22 in the third paragraph on page 1 of the manuscript as shown below:

"This represents a major technological challenge for conventional periodic poling methods (typically limited to periods $>2 \mu\text{m}^{19,22,23}$)..."

e) "ABC -type multilayers " in Discussion paragraph 1 , please clarify what this means

This term refers to recent work (original Refs. 28-30) focusing on an analogous approach to the DAS strategy adopted here, but with inorganic oxide-based materials. In this case, an effective bulk $\chi^{(2)}$ response is realized from three different (centrosymmetric) materials by layering them (materials A, B, and C) in a non-centrosymmetric ...ABCABCABCABC... stack so that their inherent interface nonlinearities do not cancel. Our use of the term 'ABC' is simply an attempt to connect readers with the terminology used in those papers.

To clarify this point, we have modified the text on page 10 to read:

"While the $\chi^{(2)}$ nonlinearity of the DAS systems studied here compares favorably to that obtained recently for similar, inorganic ABC-type multilayers (where non-cancelling interfacial nonlinearities between three different oxide layers combine to enable $d_{33}^b \sim 0.2 - 6 \text{ pm/V}^{29-31}$,..."

REVIEWERS' COMMENTS:

Reviewer #1 (Remarks to the Author):

The authors have answered in a satisfactory manner to my comments and objections. I therefore deem their manuscript acceptable for publication as is.

Reviewer #2 (Remarks to the Author):

The authors have satisfactorily addressed the numerous comments raised by the referee reports. Appropriate comments and references are included in the revision, and additional linear characterisation included in the SI. In my opinion, the paper is in a suitable form for publication.

Reviewer #3 (Remarks to the Author):

The authors appear to have addressed all the concerns and comments of the reviewers. I recommend that the manuscript be published.